# Implementing a new clinical pathway in a non-receptive context: Mixed methods evaluation of a new fracture pathway for older people in a hospital Trust in the West Midlands, UK

**Gill Combes[1], Gareth Owen[2], Sarah Damery [1]\*, Sarah Flanagan[1], Celia Brown[3], Graeme Currie[2]**

1 Institute of Applied Health Research, University of Birmingham, Edgbaston, Birmingham, United Kingdom, 2 Warwick Business School, The University of Warwick, Coventry, United Kingdom, 3 Warwick Medical School, The University of Warwick, Coventry, United Kingdom

\* s.l.damery@bham.ac.uk

**Data Availability Statement:** Study data cannot be publicly shared even if de-identified due to concerns over participant confidentiality and

## Abstract

### Objectives

This paper reports a mixed methods evaluation of a new pathway to improve clinical outcomes for older people with fractures treated at a hospital Trust in the West Midlands, UK. The paper focuses specifically on the context surrounding the translation of the new pathway into practice and the way that external and internal factors influenced its adaptation and implementation.

### Methods

Quantitative analysis used a controlled Interrupted Time Series (ITS) to estimate the effect of the new pathway on patient complication rate, median length of hospital stay and 30-day mortality by comparing the pre- and post-intervention periods. ITS data were extracted from the UK Trauma Audit and Research Network (TARN) database and a patient-level control group identified using propensity score matching. Parallel qualitative analysis aimed to examine the context surrounding the new pathway and how external and internal factors might influence its adaption and implementation into clinical practice. Data were collected via semi-structured interviews (n = 16) undertaken with staff and clinical stakeholders within the Trust and were analysed using the COM-B (Capability, Opportunity, Motivation) model of behaviour.

### Results

No statistically significant effects were found for any of the patient outcomes studied in the controlled ITS analysis. Qualitative data suggest that the lack of effectiveness of the new initiative can be explained with reference to the capability, opportunity and motivation of internal Trust stakeholders to engage with the pathway, which created a non-receptive environment within the Trust.

privacy, and due to the terms of participant consent, as noted by the REC that approved the study. Excerpts of interview transcripts relevant to the study are available on request from the research governance office of the University of Birmingham (researchgovernance@contacts. bham.ac.uk).

**Funding:** All authors were funded by the National Institute for Health Research Collaboration for Leadership in Applied Health Research and Care West Midlands (NIHR CLAHRC WM), now recommissioned as NIHR Applied Research Collaboration West Midlands (https://warwick.ac. uk/fac/sci/med/about/centres/arc-wm/). This is an infrastructure grant and has no specific grant number. The views expressed in this publication are those of the author(s) and not necessarily those of the NIHR or the Department of Health and Social Care. The funder had no role in study design, data collection and analysis, decision to publish, or preparation of the manuscript.

**Competing interests:** The authors have declared that no competing interests exist.

## Conclusions

Successfully implementing new care pathways in environments that may be non-receptive to change requires efforts to be put into winning 'hearts and minds' within the organisation to ensure engagement from key stakeholders during intervention development. Evidence must be provided internally of the way that a given intervention will alleviate the problematic issues being experienced within the organisation, and external dissemination of results should be avoided until there is evidence of a positive effect within the organisation where the new care pathway is first implemented.

## Introduction

Innovation that aims to improve healthcare delivery involves complex socio-behavioural processes [1]. Such interventions are likely to be highly context-dependent [2], and understanding the context in which change takes place is central to the success of implementation. Contexts may be 'receptive' or 'non-receptive' to change [3]. Receptive contexts are commonly defined as situations that 'seem to be favourably associated with forward movement' [3], whereas non-receptive contexts are situations where a combination of conditions 'effectively create blockages or resistance to change' [4]. Non-receptive contexts hinder diffusion of new innovation [5], and can be associated with a range of interpersonal factors such as the emotional capability of stakeholders to engage with change [6], or organisational factors such as the procedures for communication across organisational boundaries [7].

Numerous models exist for measuring behavioural change [8]. Michie et al.'s COM-B change model–developed following a systematic review of 195 behavioural change interventions–has been used widely to evaluate behaviour change in a range of health and social care settings. The COM-B framework allows consideration of the psychological, organisational and system-level factors supporting behaviour change in order to sustain, spread and scale up evidence-based innovation, by highlighting the importance of enhancing capability, opportunity and motivation to adopt new practice [9]. In this paper, we examine the context and behaviours of stakeholders from an evaluation of the implementation and effectiveness of a new trauma management pathway introduced to a large West Midlands hospital. We use Michie et al.'s COM-B model as our framework, and: a) consider how the capability, motivation and opportunity of stakeholders in the organisation to engage with the new pathway impacts on its effectiveness, and b) evaluate the impact of the receptiveness of the organisational context on the success of the intervention.

### The intervention

The intervention evaluated in this manuscript is a new clinical pathway called HECTOR (**H**eartlands **E**lderly **C**are, **T**rauma & **O**ngoing **R**ecovery), developed by the trauma lead of a large West Midlands hospital. The older population ($\geq$ 65 years), is particularly vulnerable to trauma, and often experiences poor survival when faced with multiple injuries [10]. The likelihood of survival in the older population is further complicated by comorbidities, polypharmacy and pre-morbid status, all of which influence an individual's response to trauma and may influence the clinician's attitude towards patient management [11]. Furthermore, older patients are less able to respond to traumatic injury [12] and have increased levels of trauma-related mortality [13].

The HECTOR pathway was underpinned by the broad concepts of holistic and 'joined up' care for older patients outlined within the British Geriatrics Society 'Silver Book' [14] and in the NICE guidelines for the management of delirium [15]. It was developed partially as a response to the changing demographic of patients admitted to the trauma unit at the hospital (increasing age, frailty and comorbidities, resulting in increased lehgth of hospital stay due to severe injury patterns requiring complex clinical management), and partially following the success of a recently introduced national neck of femur fracture pathway that had improved outcomes for older patients.

The HECTOR pathway aimed to improve outcomes for older people (aged 65+) with fractures treated in Accident & Emergency (A&E) and acute care, by delivering care focused on the holistic needs of the patient rather than just the presenting injury. In A&E, existing triage methods were supplemented with enhanced primary and secondary surveys and rapid imaging access. A new screening tool, the 'silver survey', aimed to identify possible medical causes for injuries and risk factors for delirium and other complications. For patients admitted to wards, a new daily assessment tool for use by the multidisciplinary team was introduced, aimed at proactively preventing and managing complications. This consisted of a short checklist for staff to assess hydration; eating and toileting; confusion and mental state; thromboprophylaxis; occult illness, and progress towards recovery. A HECTOR staff training programme was developed and implemented from September 2014 for staff across all disciplines involved in delivering the pathway.

## The evaluation

We undertook a mixed methods evaluation of the effectiveness and implementation of the HECTOR pathway. The evaluation team was independent from the clinical team at the hospital, and played no role in the design, development or implementation of the HECTOR pathway itself. Instead, the evaluation aimed to assess both patient and process outcomes using both quantitative and qualitative approaches. Quantitative analysis assessed the effectiveness of the new pathway by exploring impacts on patient outcomes (complication rates, 30 day mortality and length of hospital stay) by comparing outcomes before and after the change in care delivery.

In parallel, semi-structured interviews with staff and stakeholders at the Trust aimed to understand their experiences of managing and delivering the pathway, the context and climate within which the pathway was introduced, and how internal and external forces may have affected pathway implementation and effectiveness. As the new pathway involved both the adaptation of existing clinical practices (e.g. fast-tracked imaging) and the introduction of some new practices (e.g. the silver survey), the evaluation used the COM-B analytical framework [9] to explore how effectiveness was affected/mediated by the context in which the pathway was implemented.

## Materials and methods

The study was approved by the NRES North West Greater Manchester West Research Ethics Committee (Ref: 14/NW/1496). Approval was also obtained from the research governance office of the participating hospital Trust.

### Quantitative data

The aim of the quantitative analysis was to assess the effectiveness of HECTOR on three patient-level outcomes: the primary outcome was complication rate (the proportion of patients with one or more of seven complications: *C Difficile*, MRSA, bronchopneumonia/lower-

respiratory tract infection, urinary tract infection, pulmonary embolism, deep vein thrombosis (DVT) and pressure sore). Median length of hospital stay and 30-day mortality are also reported. A controlled interrupted time-series analysis (ITS) was undertaken allowing the pre- and post-intervention periods to be compared, along with an external control group populated using matched data from the UK Trauma Audit and Research Network (TARN) database. Months were chosen as the unit of analysis, comparing a 32 month pre-intervention period (01/01/2012 to 31/08/2014) with a 40 month post-intervention period (01/09/2014 to 31/12/2017). Control patient data were obtained from all 152 trauma units in England, extracted from the UK Trauma Audit and Research Network (TARN) database (Box 1).

## Box 1. The Trauma Audit and Research Network (TARN)

The Trauma Audit & Research Network (TARN) is a collaboration of hospitals from England, Wales, Ireland and parts of Europe. TARN collects data from all hospital Trusts via an online recording system. Cases are created and given a unique reference number and data are entered in a specific format (https://www.tarn.ac.uk/content/downloads/ 53/Procedures%20manual.pdf). For each episode, data are collected with reference to the pre-hospital episode, Emergency Department care (including imaging and procedures undertaken), operative management, critical care management and ward-based care. Information is collected regarding discharge destination, inpatient complications and TARN themselves generate Injury Severity Scores (ISS) and Probability of Survival (Ps) for each patient. In order for patients to be eligible for TARN data entry, they must meet the following criteria: be admitted for >72 hours; be transferred and then admitted for >72 hours; be admitted to critical care, irrespective of length of stay, or suffer death as an inpatient. If they meet such criteria, a patient episode is only created if they have suffered from specific types of injury (burn, facial fracture, femoral fracture, foot or toe: joint or bone, hand or finger: joint or bone, inhalation, joints, limb fracture—excluding femur, muscle, tendon or ligament, nerve, pelvis, skin, spine, vessel). For this evaluation, a further inclusion criterion was that patients had to be at least 65 years old.

A potential patient-level control group of 86,314 patients was identified, followed by propensity score matching based on nearest neighbour pair matching to identify appropriate patients for inclusion on the basis of baseline patient characteristics (age, gender, comorbidity, entrapment at scene, mechanism of injury (MOI), penetrating injury, year and month of admission). For each outcome, the ITS analysis assumed an *a priori* intervention effect which included a change in level and/or a change in slope. The "itsa" command in Stata v14 was used for the ITS analysis [16], using Newey-West standard errors and the number of patients admitted in the relevant month as the analytical weight, to reflect the variability in admissions on a month-by-month basis. As three outcomes were used, a p-value of <0.0167 was considered statistically significant.

**Statistical power.** The power to detect a change in complication rate from a baseline of 40% was estimated using a simulation over 1,000 replications of an ITS analysis with Newey-West standard errors in R, in which the standard deviation of the baseline complication rate was assumed to be 10%. Two possible outcomes were used: 1) a level effect of -5 percentage points but no further change, and 2) a level effect of -1 percentage points and a time effect to produce a further -4 percentage point change over 12 months. Given a total of 72 time periods (32 pre- and 40 post-intervention), the study was underpowered, with an estimated statistical power of 24% for the first outcome and 31% for the second (alpha = 0.05).

## Qualitative data

The qualitative evaluation aimed to examine the context surrounding the translation of the new pathway and how external and internal factors might influence its adaption and implementation into clinical practice. Semi-structured interviews were undertaken with staff and clinical stakeholders in a number of roles and at different levels of the Trust organisational hierarchy, to capture a range of perspectives on the new pathway and its adoption.

Participants were asked how they experienced HECTOR, their perceptions of the benefits and challenges, and how the new pathway influenced their work. Given the challenging back-drop to the implementation of HECTOR, the decision to use the COM-B framework to aid the evaluation's interpretation of pathway implementation was made *a priori*, and the topic guide (S1 File) was structured so that the chronology of HECTOR and its impacts could be explored, by asking participants about the development of HECTOR, its implementation, impacts on clinical practice/patients/other staff, and their views about the role of the organisation and leadership in establishing the pathway at the Trust. Interviews took place in the hospital and lasted between 30–60 minutes. All were audio-recorded, and recordings were transcribed verbatim, with transcripts proof-read against recordings by the member of the research team who conducted the interview. Written informed consent was obtained from all interview participants.

Data analysis began with a fine-grained reading of the data and the inductive development of themes [17, 18]. Once this was complete, the research team met to discuss the results. Interview transcripts were analysed using the key domains of the COM-B framework to understand the way respondents described their capability, opportunity and motivation to adopt the new pathway. In using this method, we engaged in deductive reasoning, linking our inductive themes with existing concepts and frameworks [19].

# Results

## Quantitative results

**Patient characteristics.** TARN data were available for 876 patients admitted to the Trust. Propensity score matching produced two groups of 746 each (matching rate 85%) with well-balanced baseline covariates (Table 1). However, there was evidence that patients at the Trust had more severe injuries in the pre-intervention period than control patients.

**Impact of HECTOR on complications, length of stay and mortality.** Table 2 presents the results of the controlled ITS. The predicted proportion of patients with one or more complications was increasing at around the same rate in the pre-intervention period at the Trust and control hospitals, although the rate at the start of data collection was higher at the Trust. The intervention appears to have had an initial 'level' effect in reducing the complications rate at the Trust (when it increased slightly at control hospitals) and to have reversed the pre-intervention upward trend (at a faster rate than at control hospitals), although these results were not statistically significant.

Predicted length of stay was decreasing (non-significantly) in the pre-intervention period at a similar rate in both the Trust and control hospitals. While there was a small 'level' effect of the intervention at control hospitals, the equivalent effect at the Trust was smaller, although none of these effects were statistically significant. In the post-intervention period, the predicted length of stay at control hospitals decreased by 0.2 days per month, which was statistically significant, whilst the monthly decrease at the Trust was much smaller, such that the length of stay achieved at the point of the intervention was almost unchanged in the post-intervention period.

**Table 1. Descriptive statistics by time period and group after propensity score matching.**

| Variable | Pre-intervention | | Post-intervention | | Univariate comparison (pre-intervention)[a] |
|---|---|---|---|---|---|
| | Control | Trust | Control | Trust | |
| Dates | 01/01/2012 to 31/08/2014 | | 01/09/2014 to 31/12/2017 | | |
| Number of patients | 237 | 245 | 509 | 501 | |
| **CHARACTERISTICS** | | | | | |
| Mean age (SD) | 81.6 (8.2) | 81.8 (8.3) | 81.4 (8.4) | 81.9 (8.4) | t = 0.24; p = 0.81 |
| Gender: N (%) male | 71 (30.0) | 87 (35.5) | 197 (38.7) | 189 (37.7) | $\chi^2$ = 1.69; p = 0.194 |
| Injury Severity Score: median (IQR)[b] | 9 (9 to 13) | 10 (9 to 16) | 9 (9 to 16) | 9 (9 to 17) | Bonnet-Price median **p<0.0001** |
| Mode of injury: N (%) | | | | | |
| Fall <2m | 200 (84.4) | 197 (80.4) | 430 (84.5) | 418 (83.4) | |
| Fall >2m | 21 (8.9) | 25 (10.2) | 54 (10.6) | 51 (10.2) | $\chi^2$ = 1.54; p = 0.672 |
| Vehicle incident | 13 (5.5) | 18 (7.4) | 20 (3.9) | 26 (5.2) | |
| Other | 3 (1.3) | 5 (2.0) | 5 (1.0) | 6 (1.2) | |
| Most severely injured body region: N (%) | | | | | |
| Chest | 30 (12.7) | 32 (13.1) | 53 (10.4) | 57 (11.4) | |
| Head | 36 (15.2) | 46 (18.8) | 120 (23.4) | 123 (24.6) | |
| Limbs | 102 (43.0) | 95 (38.8) | 199 (39.1) | 192 (38.2) | $\chi^2$ = 2.63; p = 0.757 |
| Spine | 49 (20.7) | 50 (20.4) | 95 (18.7) | 100 (20.0) | |
| Multiple | 19 (8.0) | 22 (9.0) | 38 (7.5) | 27 (5.4) | |
| Other | 1 (0.4) | 0 | 2 (0.4) | 2 (0.4) | |
| **OUTCOMES** | | | | | |
| Median LoS[c] (days): (IQR) | 14 (7 to 27) | 15 (7 to 31) | 12 (7 to 21) | 14 (8 to 25) | |
| 1+ complications: N (%) | 23 (9.7) | 51 (20.8) | 53 (10.4) | 44 (8.8) | |
| 30 day mortality: N (%) | 12 (5.1) | 37 (15.1) | 68 (13.4) | 70 (14.0) | |

[a]Bold text indicates statistically significant result at p<0.0167

[b]IQR = Inter-quartile range

[c]LoS = length of stay.

Predicted mortality was decreasing slightly in the pre-intervention period at the Trust but was not changing at control hospitals. The only statistically significant result was a 'level' increase in the mortality rate at control hospitals of just over 10 percentage points. The 'level'

**Table 2. ITS comparison of trends in patient outcomes over time and between groups.**

| | Proportion with 1 + complication(s)[a] | Median length of stay (days) | 30-day mortality |
|---|---|---|---|
| Intercept (outcome at baseline for Control group) | 0.076 (0.075) | **17.89 (<0.001)** | 0.050 (0.147) |
| Difference in intercepts at baseline (T−Control)[b] | 0.103 (0.085) | 3.306 (0.418) | 0.130 (0.027) |
| Time effect pre-intervention in Control group (effect of each additional month from beginning of data collection) | 0.001 (0.604) | -0.123 (0.233) | 0.000 (0.979) |
| Difference in time effect pre-intervention (T−Control) | 0.001 (0.875) | -0.061 (0.755) | -0.002 (0.534) |
| Level effect of intervention in Control group (immediate effect) | 0.029 (0.612) | 3.204 (0.146) | **0.104 (0.011)** |
| Difference in the level effect immediately after intervention (T−Control) | -0.099 (0.214) | -2.602 (0.477) | -0.061 (0.329) |
| Time effect post-intervention in Control group (effect of each additional month from intervention) | -0.002 (0.075) | **-0.202 (<0.001)** | -0.002 (0.063) |
| Difference in time effect post-intervention in Control group (T−Control) | -0.002 (0.357) | 0.172 (0.018) | 0.001 (0.620) |

[a]Results shown as co-efficient and p values with significant values (<0.0167) shown in bold

[b]T = The Trust.

increase at the Trust was lower than for the controls, but not significantly so, yet it is plausible that HECTOR 'prevented' a larger increase. There was a subsequent downward trend in mortality at both the Trust and the control hospitals. However, predicted mortality remained higher at the Trust compared to control hospitals throughout the post-intervention period.

## Qualitative results

Sixteen interviews were undertaken between August and December 2014. Interviewee roles included a Trust Medical Director; Clinical Directors and Senior Emergency Medicine management (n = 3); Consultants in Emergency Medicine (n = 5), Trauma and Orthopaedics (n = 1) and Elderly Medicine (n = 2); senior nursing staff (n = 2) and data support officers (n = 2). The quantitative evaluation of the HECTOR pathway indicated no significant change as a result of implementation. The qualitative data indicated that the observed lack of effectiveness of the new pathway may have been negatively impacted by certain stakeholder behaviours at the Trust in which the pathway was introduced. Themes are presented in Table 3, organised around the COM-B framework domains of capability, opportunity and motivation for individuals to engage with the new pathway [9].

**Capability.** The physical capability of colleagues across clinical areas within the Trust to engage with the new pathway limited its success. One consultant reflected:

> *"At the end of the day, I can be 15 minutes with this one problem but another 120 problems I'm not giving my time to, so it's working out, prioritising what I need to do."* (HEC7)

Physical capability was further impaired by the entrenchment of working patterns that existed within the organisation and a lack of holistic working between trauma/orthopaedics and other clinical areas. One nurse commented:

> *"As far as they're concerned once they've fixed that leg, whether it's a plaster on, whether it's a nailing, whatever they've done, as far as they've concerned that's that bit, they're finished."* (HEC16)

**Opportunity.** Social and physical opportunities were created to encourage buy-in to HECTOR. Social opportunities were attempted through top down efforts to change the culture of the organisation. These efforts involved a series of strategic meetings to discuss how

**Table 3. Internal and external behaviours of stakeholder engagement with pathway.**

| | |
|---|---|
| Capability | Physical capability of staff to change behaviour impeded by workload and entrenched patterns of siloed working |
| | Change to certain staff behaviour impeded by lack of psychological capability of adapting to the new way of working |
| | Adoption of new pathway limited by psychological capability based on lack of engagement with the organisation |
| Opportunity | Physical opportunity evidenced by the lack of engagement with the training and publicity of the pathway amongst staff beyond A&E department |
| | Social opportunity evidenced by the lack of engagement strategically to embed the new pathway in the organisation's working practices |
| Motivation | Automatic motivation amongst colleagues within the organisation impeded by concern over risks presented by introducing new way of working |
| | Reflective motivation within the organisation impeded by concerns from colleagues about external activities of project lead |

HECTOR would be embedded across the Trust. However, these meetings were not well attended. One senior executive reflected:

> *"There was minimal response to the first meeting: when we had the first meeting, which had gone out in my name–I was pulling all the power I got, A&E turned up but T&O weren't there and Elderly wasn't there. So I wrote to them all afterwards and said: 'Come on guys, step up a bit here!'"* (HEC2)

Attempts to create physical opportunities were bottom-up and evidenced through general teaching sessions at the Trust. The teaching was supplemented by posters placed around the ward summarising the HECTOR principles:

> *"We started to put posters around the whole place, like near the site office where the bed management team worked, a few posters around. I just wanted to publicise what we're doing to make people think 'What is HECTOR?' So they would ask."* (HEC14)

It appeared that these measures to present the project to other departments beyond A&E went largely unacknowledged. However, outside the Trust, HECTOR was seen as a positive, innovative and effective pathway. The project lead was invited to lead training sessions across the West Midlands trauma network, whose members were enthusiastic and receptive. At the time of writing, The Royal College of Emergency Medicine had begun the process of adopting HECTOR into national guidelines, and the new pathway had won awards from the BMJ (2018 winner in the category of Emergency) and West Midlands Clinical Research Network (2017 winner in the category of Improvement project of the year) as well as being a NHS70 Parliamentary Award Winner in 2018 in the category of excellence in urgent and emergency care.

The project lead also used social media to promote the new pathway. HECTOR was followed on Twitter by over 1000 individuals. Using social media to engage with HECTOR provided a social opportunity for individuals to engage with the HECTOR pathway but there appeared to be minimal engagement from colleagues within the organisation.

**Motivation.** Internal motivation to engage with the new pathway was limited because individuals within the Trust were under significant scrutiny during the period in which the new pathway was introduced, and as a result, the implementation of the pathway was impeded. One consultant commented:

> *"If you have paperwork which was never in the contract of* [the] *doctor to fill out, other paperwork you don't feel inclined to fill out the paperwork, or it's not filled out properly in time, and the one who fills* [in] *the paperwork is legally responsible to follow it because of the shift system this is not working."* (HEC6)

The national recognition HECTOR had gathered further affected the motivation of groups of clinicians within the Trust to engage with the new pathway. One consultant suggested that the pathway had not been fully evaluated within the organisation:

> *"What we don't know is how good this innovation* [is] *in our own backyard rather than saying: 'We've got the national course for it'–which is fantastic but it is our backyard where it needs to put it into practice both in letter and spirit."* (HEC9)

Influential actors within the Trust felt that the new pathway should not have been promoted nationally before it had been proven successful in the host organisation. The use of social

media in particular to promote HECTOR nationally was questioned by colleagues within the Trust. The external presence of HECTOR and the perceived lack of evidence for effectiveness within the Trust meant there was less inclination amongst colleagues in Elderly Care and Trauma and Orthopaedics to engage with the new pathway. For example, one interviewee commented:

> *"I mean it's not just black or white, I suppose, you get things like internet stuff, videos, that kind of stuff. Yeah, fine, people are happy with all that kind of stuff. It's when you start getting into other things like Twitter and whatever, that kind of stuff, people are probably slightly less so."* (HEC10).

## Discussion

The quantitative evaluation of a clinical pathway to improve care for older patients admitted to the Trust with traumatic injuries aimed to examine the impact of the pathway on patient related outcomes. However, no statistically significant effects were found for any of the three patient outcomes studied in a controlled ITS analysis. Qualitative data suggest that the lack of effectiveness of the HECTOR pathway can be explained with reference to the capability, opportunity and motivation of internal Trust stakeholders to engage with the pathway, which created a non-receptive environment for HECTOR within the Trust.

### The effect of a non-receptive environment on new innovation

The HECTOR pathway was introduced at a time of considerable change for the hospital, with severe ongoing clinical and financial pressures throughout the period of the study which meant that the hospital experienced periods of extensive external scrutiny and multiple changes to senior leadership. Thus, the environment into which the new pathway was introduced was affected by a series of interrelated factors, many of which could not be readily addressed. For instance, the timing of implementation hindered the intervention because of the scrutiny the Trust was under at the time, and because of clinical specialities such as elderly care and trauma and orthopaedics historically failing to co-operate within change within the organisation. Introducing a new elderly trauma pathway at a time of organisational stress meant that attempts to change the behaviour of key stakeholders within the Trust and provide a receptive environment for pathway implementation were unsuccessful.

As is often the case with changes to clinical pathways, if the context and climate are not conducive at the development phase, implementation of the pathway will face challenges, particularly if the innovation requires radical behavioural change [20]. In the case of the HECTOR pathway, qualitative data indicate that a non-receptive environment existed in the development phase due to the behaviours of stakeholders in the organisation. Some of these could have been managed, but others represented the influence of institutionalised organisational routines which were beyond the abilities of the organisation or the project lead to change [21].

Engagement with the pathway by clinicians within the Trust was limited to those who worked closely with the project lead from the beginning of the project in the A&E department. These individuals were able to change their behaviour more readily to adopt the new way of working. Clinicians from other specialities did not engage or change their behaviours. Their lack of capability to do so was affected by the scrutiny the hospital was under and their own work pressures, as well as a lack of physical capability (i.e. time pressures) and psychological capability (i.e. entrenched working patterns). This reflects Michie et al.'s model that

behavioural change is effected by the physical and psychological capability of stakeholders to adopt the new behaviours to create a receptive environment for the innovation [9].

As a result of the unreceptive environment within the hospital, the project lead turned to external networks who were more receptive to the new pathway as they recognised the benefits of the intervention and were more enthusiastic and motivated about the pathway's potential benefits. The project lead therefore focused energy on providing opportunities for external stakeholders to engage with the new pathway and attempted to effect behavioural change by constructing legitimacy for the project through stakeholders beyond the auspices of the organisation [22]. However, instead of the recognition that the new pathway had gained momentum nationally working to leverage support from within the Trust, this external recognition impacted negatively on the behaviour of internal stakeholders who became less motivated to engage with the new pathway and who perceived its benefits as remaining unproven within their local context. As the new pathway gathered support externally, clinicians from other specialities within the organisation invoked their professional status to justify distancing themselves from the new innovation [23] as further attempts to present opportunity for engagement with the new pathway across the organisation were made internally.

## The utility of mixed methods to examine results from complex clinical interventions

In any mixed methods study, the purpose of mixing qualitative and quantitative methods should be clear in order to determine how they relate to one another and how the data should be integrated [24]. Mixed methods studies tend to attempt triangulation approaches to achieve empirical clarity, or theory building through aligning results from mixed methods data collection [25, 26].

Our study highlighted the particular value of comparing the qualitative and quantitative data at the interpretative stage of the research process [27], as this allowed a deeper understanding of the organisational context in order to evaluate factors that influenced the observed quantitative finding with regard to the patient outcomes assessed. In doing so, this led to the conclusion at the interpretive stage that the unreceptive organisational environment had influenced the degree to which the new pathway was able to demonstrate effectiveness. Although we cannot prove causality regarding the non-receptive environment as the reason for the counter-intuitive results of the intervention on patient outcomes, our mixed methods approach does highlight the value of considering the wider contexts of evaluations of new interventions, an area which continues to be of much debate [28, 29]. Through applying a mixed methods approach that focused on wider contextual explanations provided by the qualitative data at the interpretive stage of the research process we were able to offer contextual explanations of our results that highlighted the central role that a unreceptive organisational environment plays in the success (or lack thereof) of a new innovation.

## Implications for practice

The challenges of translating a behaviour change intervention into a non-receptive context identified in this evaluation suggest several implications for others who may be attempting similar interventions within their own organisations. First, it is clear that substantial efforts need to be put into winning 'hearts and minds' within the organisation that will be host to any new intervention, since effective implementation requires 'buy-in' and engagement from staff at all levels of seniority and in all roles. Second, evidence needs to be provided internally of the way that any proposed intervention may help to alleviate issues that are contributing to problems within the organisation (e.g. evidence that the new pathway will help to save money or

improve outcomes that the Trust is being monitored on). This may facilitate the necessary engagement from staff to implement the intervention effectively. And third, external dissemination of the results from the QI intervention should be avoided until there is evidence that the intervention has a positive effect on outcomes in the place where it is first implemented, otherwise the goodwill of clinical and nursing staff within the host institution may dissipate.

## Limitations

There was some evidence of under-reporting to the TARN database. Annual case ascertainment rates ranged from 53% (2013) to 75% (2012) for the evaluation site, and from 60.3% (2012) to 84.2% (2017) in control hospitals. If the under-reporting was systematic (e.g. the most seriously injured patients were not included), this could have introduced bias into the results. The ITS was also substantially underpowered to detect differences in outcomes over time. There were also some differences in the characteristics of patients who were and were not matched between the Trust and control hospitals: the matching rate was higher for males, younger patients and those with limb injuries, and lower for those with injuries caused by a fall of less than 2m and with head or spinal injuries. There was some evidence that patients at the Trust had more severe injuries in the pre-intervention period than control patients, which could explain the differences in outcomes between the two groups in this period (higher complication rates, length of stay and 30-day mortality at the Trust compared to control pre-intervention). Nevertheless, the matching process did produce well-balanced groups and enabled a controlled ITS analysis to be undertaken, which was the most robust approach possible in the absence of a randomised controlled trial design.

## Conclusions

Our evaluation indicates two broad conclusions. First, the importance of considering the effects of non-receptive environments on the success of new interventions. Second, an acknowledgement of the value of employing mixed methods and using qualitative data within evaluations of QI interventions to help provide explanations at the interpretive stage of the research process. This allows evaluation to consider the wider contextual factors associated with intervention development and implementation that might influence the observed effects, over and above those indicated purely by clinical outcomes.

## Supporting information

**S1 File. Topic guide for semi-structured interviews.**
(PDF)

## Acknowledgments

A patient advisory group contributed to the initial design of the study and the data collection tools, alongside the patient and public involvement group of the long term conditions theme of CLAHRC-WM.

## Author Contributions

**Conceptualization:** Gill Combes, Graeme Currie.

**Data curation:** Gill Combes, Gareth Owen, Sarah Damery, Sarah Flanagan.

**Formal analysis:** Gill Combes, Gareth Owen, Sarah Damery, Sarah Flanagan, Celia Brown, Graeme Currie.

**Methodology:** Gill Combes, Gareth Owen, Sarah Damery, Sarah Flanagan, Celia Brown.

**Supervision:** Gill Combes, Graeme Currie.

**Validation:** Gill Combes, Gareth Owen, Sarah Damery, Sarah Flanagan, Celia Brown, Graeme Currie.

**Writing – original draft:** Gill Combes.

**Writing – review & editing:** Gareth Owen, Sarah Damery, Sarah Flanagan, Celia Brown, Graeme Currie.

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
