## [Decision Letter · Decision Letter 0]

18 Nov 2020

PONE-D-20-06089

Implementing a quality improvement intervention in a non-receptive context: case study of a new fracture pathway for older people in a hospital Trust in the West Midlands, UK

PLOS ONE

Dear Dr. Damery,

Thank you for submitting your manuscript to PLOS ONE. After careful consideration, we feel that it has merit but does not fully meet PLOS ONE’s publication criteria as it currently stands. Therefore, we invite you to submit a revised version of the manuscript that addresses the points raised during the review process.

We look forward to receiving your revised manuscript.

Kind regards,

Kathleen Finlayson

Academic Editor

PLOS ONE

Journal Requirements:

2. Please include additional information regarding the interview guide used in the study and ensure that you have provided sufficient details that others could replicate the analyses. For instance, if you developed a guide as part of this study and it is not under a copyright more restrictive than CC-BY, please include a copy, in both the original language and English, as Supporting Information.

3.We note that you have indicated that data from this study are available upon request. PLOS only allows data to be available upon request if there are legal or ethical restrictions on sharing data publicly. For information on unacceptable data access restrictions, please see http://journals.plos.org/plosone/s/data-availability#loc-unacceptable-data-access-restrictions.

Additional Editor Comments (if provided):

Thank you for your submission, the topic is of significance and the study results valuable in building knowledge of this area.

There are a couple of areas where further explanation would improve understanding of your methods, i.e., re the change model used to evaluate the behaviour change - has this model been used and validated in previous work? Was the model used to design the implementation of your intervention? I note you say it was used after the end of the study, thus did you use another model to guide implementation of the pathway?

Re the HECTOR pathway - how was this developed? was a review of the literature undertaken? were the staff in the setting of this study involved in developing the pathway?

The wording of the last sentence in your sample size power calculation is confusing (to a non-statistician). Do you mean you only had power of 24 /31%? or is this the expected change in complication rates?

The manuscript is well written and the results explained clearly.

Please note further suggestions from the reviewers below,

Reviewers' comments:

Reviewer's Responses to Questions

**Comments to the Author**

1. Is the manuscript technically sound, and do the data support the conclusions?

Reviewer #1: Yes

2. Has the statistical analysis been performed appropriately and rigorously? 

Reviewer #1: Yes

3. Have the authors made all data underlying the findings in their manuscript fully available?

Reviewer #1: No

4. Is the manuscript presented in an intelligible fashion and written in standard English?

Reviewer #1: Yes

5. Review Comments to the Author

Reviewer #1: RELEVANCE AND ORIGINALITY

With the growing acknowledgement of the importance of context on successful and sustainable implementation of evidence-based innovations, this manuscript is both important and relevant. The manuscript presents findings from a mixed methods study exploring possible reasons for failed uptake of an intervention at a local site (internal) despite widespread recognition and update (external). The manuscript is original in its presentation by not only presenting the findings from the study but also discussing the value of mixed methods studies to help explain why an intervention worked or not, using the format of a case study.

AUTHENTICITY AND REFERENCING

The manuscript appears to be the work of the authors and gives appropriate attribution to the work of other authors in the in-text citations and the reference list.

TITLE, ABSTRACT, AIM

The title is informative and highlights the main idea or focus of the manuscript. The abstract is structured and contains key information about the original study. Although there are objectives for the manuscript presented in the abstract, there are no aims, objectives or research questions explicitly identified for the original study in the manuscript. The reader is left to decide for themselves the authors’ intent in this regard, which is risky for misinterpretation. The study is reported to be a ‘single site’, ‘mixed methods’ , ‘case-study evaluation’ under the banner of ‘quality improvement’ employing a ‘controlled interrupted time series’ (quantitative) design and ‘semi structured interview’ (qualitative) data collection methods. This probably should be unpacked better in the methodology section.

INTRODUCTION

The background should build up to the gap in the literature followed by a clear aim or objective of the study. There is no background related to the population or intervention (ie care pathways / fractures presenting to A&E) and limited background on what seems to be the focus of the paper – a non-receptive context and a framework for ‘measuring’ behavioural change. However, as the objective is to present a case study of evaluation of a non-receptive environment, this may be appropriate if journal guidelines require a brief introduction section. It seems as though the environment was already determined to be non-receptive. The chronology of key events needs clarification – implementation of innovation, innovation deemed ineffective, context determined to be non-receptive, evaluation of determinants of non-receptive contexts … or was context determined to be non-receptive after the evaluation of the semi structured interviews?

RESEARCH DESIGN, METHODS, ANALYSIS

As stated earlier - the study is reported to be a mixed methods (METHODOLOGY), wrapped up in a case-study (DESIGN FRAMEWORK) of an evaluation (DESIGN FRAMEWORK), under the banner of ‘quality improvement’ (DESIGN FRAMEWORK) employing a ‘controlled interrupted time series’ (QUANTITATIVE DESIGN FRAMEWORK) design and ‘semi structured interview’ (QUALITATIVE DATA COLLECTION METHOD). This requires unpacking, organisation and explanation for clarity in this methodology section. The last sentence of “The Evaluation” section requires re-wording for clarity (page 4, lines 86-89). Details for quantitative and qualitative data collection and analysis are sufficient. Re statistical power, the authors estimated power of 24% and 31%, for complication rate and hospital length of stay, respectively. This seems low relative to the convention of 80% power. If I am incorrectly interpreting the intent of these figures, I’m happy to be corrected. In any case, an explanation is needed.

RESULTS, DISCUSSION, CONCLUSION

The results are generally clearly presented, in a few places there is some interpretation mixed in with results. For example,

• page 7, line 171 – “…which could explain…”

• page 8, lines 198-199 – ”…This was unexpected given the lack of any intervention and the 198 reason for this result is unclear.”

The discussion and recommendations are supported by the findings presented.

TABLES AND FIGURES

Data presented in Tables 1, 2 and 3 are clear. Caption for Table 2 could be improved by making it more meaningful.

WRITING STYLE

The writing style is unencumbered and concise. Generally easy to read.

OTHER

The others mention several times of ‘other initiatives affecting UK hospitals (page 4, lines103-104. IT would be helpful to explain /describe these other initiatives and provide some supporting evidence for the claims.

The subheading, Controlled interrupted time-series analysis (ITS), could be replaced with a more meaningful subheading eg “Impact of HECTOR on complications, length of stay and mortality”

SCHOLARLY APPROACH

A scholarly approach begins with a clearly stated premise so that compelling arguments can be presented and supported with a critical analysis of relevant, up to date literature, including empirical research evidence. The background is limited; however, a case for the evaluation can be seen, although this could be strengthened by some critique of the previous literature.

OVERALL COMMENTS

I commend the authors on undertaking their study with the intention of disseminating the findings wider. My comments have been provided in the spirit of collegiality to hopefully assist the authors in improving the manuscript.

6. PLOS authors have the option to publish the peer review history of their article (what does this mean?). If published, this will include your full peer review and any attached files.

Reviewer #1: No

---

## [Author Response · Author response to Decision Letter 0]

12 Jan 2021

PONE-D-20-06089

Implementing a quality improvement intervention in a non-receptive context: case study of a new fracture pathway for older people in a hospital Trust in the West Midlands, UK

We would like to thank the editor and reviewer for their comments on the paper and for the opportunity to submit a revised version responding to the feedback received. This response sets out each of the comments (which have been numbered for ease of distinguishing between them), and describes how we have responded.

Editorial comments:

1. Please ensure that your manuscript meets PLOS ONE’s style requirements, including those for file naming.

Response: We have carefully checked the manuscript and author affiliations against the journal criteria and have made any formatting changes required.

2. Please include additional information regarding the interview guide used in the study and ensure that you have provided sufficient details that others could replicate the analyses. For instance, if you developed a guide as part of this study and it is not under copyright more restrictive than CC-BY, please include a copy, in both the original language and English, as Supporting Information.

Response: We have included a copy of our interview topic guide as supporting information (S1_File.pdf). A clearer description of the structure of the topic guide has been added to page 6 of the manuscript under the ‘qualitative data’ heading (lines 166-169).

3. We note that you have indicated that data from this study are available upon request. PLOS only allows data to be available upon request if there are legal or ethical restrictions on sharing data publicly. In your revised cover letter, please address the following prompts:

a) if there are ethical or legal restrictions on sharing a de-identified data set, please explain them in detail. Please also provide contact information for a data access committee, ethics committee or other institutional body to which data requests may be sent. 

b) If there are no restrictions, please upload the minimal anonymized data set necessary to replicate your study findings.

Response: There are ethical restrictions, imposed by the REC who approved the study, which mean that we cannot share a de-identified data set. These primarily relate to concerns around the identification of participants given that these were senior trauma and orthopaedic staff at a hospital in the West Midlands. As a result, we would like our data availability statement to read as follows:

“Study data cannot be publicly shared even if de-identified due to concerns over participant confidentiality and privacy, and due to the terms of participant consent, as noted by the REC that approved the study. Excerpts of interview transcripts relevant to the study are available on request from the research governance office of the University of Birmingham (researchgovernance@contacts. bham.ac.uk)”. 

4. There are a couple of areas where further explanation would improve understanding of your methods i.e. re the change model used to evaluate the behaviour change – has this model been used and validated in previous work? Was the model used to design the implementation of your intervention? I note you say it was used after the end of the study, thus did you use another model to guide implementation of the pathway?

Response: Thank you for this point. The COM-B model has been widely used to assess the implementation of healthcare interventions in many settings. Some additional text has been added to the introduction to describe the rationale for using COM-B as a framework, and also to the methods (qualitative section) to explain further how the COM-B framework was actually applied to the qualitative data.

The model was not used to design the implementation of the intervention – the intervention was designed and implemented by the clinical team at the Trust and we (the authors) played no role in design or implementation of the new pathway. Our role as academic partners was to undertake an independent evaluation of the effectiveness of the intervention, and to evaluate the strengths and weaknesses of the implementation of HECTOR as a pathway within the hospital. Some additional explanatory text has been added to the ‘evaluation’ section of the introduction to clarify the respective roles of the clinical team and the independent academic team undertaking the evaluation. 

5. Re the HECTOR pathway – how was this developed? Was a review of the literature undertaken? Were the staff in the setting of this study involved in developing the pathway?

Response: We have added some more background to the HECTOR pathway and the values/rationale underpinning its development. This has been placed within the ‘intervention’ section (pages 3-4) and describes the clinical reasons that drove the decision to develop the new pathway, as well as the concepts underpinning its design. The pathway was designed by the trauma lead at the hospital, who took the lead in driving forward the changes to clinical practice that the new pathway entailed. As stated in our response to point 4 above, the evaluation team had no role in the design of the pathway or its implementation. 

6. The wording of the last sentence in your sample size power calculation is confusing (to a non-statistician). Do you mean you only have power of 24/31%? Or is this the expected change in complication rates?

Response: Apologies for confusion. The power was indeed only 24/31% at an alpha of 0.05, and the study was underpowered to detect significant changes in patient outcomes when the pre- and post-HECTOR time periods were compared. The recognition that the study was underpowered is noted in the statistical power section (page 6) and has been added to the limitations section (page 16). 

Reviewer comments:

1. [Title, abstract, aim] Although there are objectives for the manuscript presented in the abstract, there are no aims, objectives or research questions explicitly identified for the original study in the manuscript. The reader is left to decide for themselves the authors’ intent in this regard, which is risky for misinterpretation. 

Response: Thank you for this point. The objectives for the manuscript are those that underpinned the evaluation itself i.e. to assess the effectiveness of the new pathway on patient outcomes (quantitative analysis), and to understand the experiences of staff and stakeholders involved in delivering the new pathway, the context within which it was introduced, and how context may have mediated/affected effectiveness (qualitative analysis). We believe that the changes made to the ‘intervention’ and ‘evaluation’ sections on pages 3 and 4 make the aims of the evaluation (and thus the manuscript) clearer, and in particular, that this makes the distinction between what was done by the clinical team and by the independent evaluation team clearer also.

We feel that having an additional standalone aim/objectives section would be unduly repetitive (although we would be happy to add this if directed to do so), but by the same token, removing the aims from the ‘intervention’ and ‘evaluation’ sections to create an aims/objectives section would in turn make these sections harder to understand. However, the aim of the quantitative and qualitative elements of the evaluation are reiterated in the methods at the start of their respective sections (pages 5 and 7). 

2. [Title, abstract, aim] The study is reported to be a ‘single site’, ‘mixed methods’, ‘case-study evaluation’ under the banner of ‘quality improvement’ employing a ‘controlled interrupted time series’ (quantitative) design and ‘semi-structured interview’ (qualitative) data collection methods, This probably should be unpacked better in the methodology section.

Response: We agree that there are a lot of concepts at play here! We have changed the title to simplify it whilst still capturing what the study tried to do. The single site nature of the evaluation is self-evident and described clearly in the ‘intervention’ and ‘evaluation’ sections. The fact that the study is mixed methods (and the relative aims of the quantitative and qualitative work) is hopefully more clearly articulated following the changes to the introduction and methods sections. Similarly, the fact that this is an independent evaluation of a new clinical pathway is hopefully clearer following changes made to the introduction and methods sections, along with a clear rationale for the use of ITS and semi-structured interviews. 

References to the evaluation as a ‘case study’ have been removed, as the case study label is superfluous. Similarly, framing the work under the quality improvement umbrella rather than simply labelling it as a new clinical pathway or behaviour change intervention was considered superfluous also, so references to QI have been removed. We hope that these changes make the various concepts used in the manuscript clearer. 

3. [Introduction] The background should build up to the gap in the literature followed by a clear aim or objective of the study. There is no background related to the population or interview (i.e. care pathways/fractures presenting to A&E) and limited background on what seems to be the focus of the paper – a non-receptive context and a framework for ‘measuring’ behavioural change. However, as the objective is to present a case study of evaluation of a non-receptive environment, this may be appropriate if journal guidelines require a brief introduction section. If seems as though the environment was already determined to be non-receptive. The chronology of key events needs clarification – implementation of innovation, innovation deemed ineffective, context determined to be non-receptive, evaluation of determinants of non-receptive contexts…or was context determined to be non-receptive after the evaluation of the semi-structured interviews?

Response: Thank you for this point. When drafting this manuscript, we began with a ‘conventional’ literature review, focusing on the population at risk from trauma, issues such as the ageing population with complex needs etc. However, as we went through successive drafts of the manuscript, it became clear that the population was of secondary importance to the issues relating to intervention implementation and context. So, the latter became our focus, given that the evaluation was primarily focused on identifying and understanding the factors that may influence how an intervention is put in place within a clinical context and whether or not it can be effective.

Journal guidelines require a brief introduction, which we have chosen to focus on the behaviour change aspects of the topic that we evaluated. However, we have added some background and additional citations about the elderly trauma population under the ‘intervention’ section. This additional text makes the clinical area of investigation clearer and explains why there was a clinical need for an elderly trauma like HECTOR to be developed. 

The environment was not determined to be non-receptive before the evaluation began, this only became clear once the interviews had been analysed and once we discussed the quantitative and qualitative findings alongside each other. There were a few instances in the manuscript where the text implied that the context was deemed to be non-receptive from the outset. This is incorrect and such instances have now been removed. The fact that the context was determined to be non-receptive after the qualitative data were analysed is hopefully clearer now. 

4. [Research design, methods, analysis] As stated earlier – the study is reported to be a mixed methods (METHODOLOGY), wrapped up in a case-study (DESIGN FRAMEWORK) of an evaluation (DESIGN FRAMEWORK), under the banner of ‘quality improvement’ (DESIGN FRAMEWORK) employing a ‘controlled interrupted time series’ (QUANTITATIVE DESIGN FRAMEWORK) design and ‘semi structured interview’ (QUALITATIVE DATA COLLECTION METHOD). This requires unpacking, organisation and explanation for clarity in the methodology section.

Response: See response to point 2 above where we describe how we have made changes to the manuscript to simplify our description of what we did. 

5. The last sentence of ‘The Evaluation’ section requires rewording for clarity (page 4, lines 86-89).

Response: This sentence has been reworded. 

6. Re statistical power, the authors estimated power of 24% and 31% for complication rate and hospital length of stay, respectively. This seems low relative to the convention of 80% power. If I am incorrectly interpreting the intent of these figures, I’m happy to be corrected. In any case, an explanation is needed.

Response: The reviewer is correct, and yes, this shows the quantitative work was underpowered. We have acknowledged this in the statistical power section and added a note to the limitations. 

7. [Results, discussion, conclusion] In a few places there is some interpretation mixed in with results. For example:

Page 7, line 171 “…which could explain…”

Page 8, lines 198-99 “…this was unexpected given the lack of any intervention and the reason for this result is unclear…”

Response: Thank you for pointing this out. The places in which interpretation is mixed in with results have been removed. In the case of “…which could explain…” from page 7, this text has been moved to the limitations section as it relates to the quality of the matching between Trust patients and controls. 

In the case of “…this was unexpected…”, the sentence has been removed completely. 

8. [Tables and figures] Caption for Table 2 could be improved by making it more meaningful.

Response: We agree – the caption has been changed to “ITS comparison of trends in patient outcomes over time and between groups”

9. [Other] The authors mention several times ‘other initiatives affecting UK hospitals (page 4, lines 103-4). It would be helpful to explain/describe these other initiatives and provide some supporting evidence for the claims.

Response: Thank you for pointing this out. We have no specific examples to offer, other than the fact that there is usually some new initiative being tried in every hospital to improve care, usually driven by government incentives like the Better Care Fund or the Sustainability and Transformation Partnerships in recent years to name just two. We appreciate that these statements seem vague and we cannot substantiate them appropriately, so they have been removed from the manuscript. 

10. The subheading ‘controlled interrupted time-series analysis (ITS) could be replaced with a more meaningful subheading e.g. “Impact of HECTOR on complications, length of stay and mortality”

Response: We agree, we have replaced the subheading with the text suggested by the reviewer.

---

## [Editor Report · Decision Letter 1]

8 Feb 2021

Implementing a new clinical pathway in a non-receptive context: mixed methods evaluation of a new fracture pathway for older people in a hospital Trust in the West Midlands, UK

PONE-D-20-06089R1

Dear Dr. Damery,

We’re pleased to inform you that your manuscript has been judged scientifically suitable for publication and will be formally accepted for publication once it meets all outstanding technical requirements.

Kind regards,

Kathleen Finlayson

Academic Editor

PLOS ONE

Additional Editor Comments (optional):

Thank you for addressing the reviewers' comments, the methods of the study are much clearer.
---

## [Editor Report · Acceptance letter]

11 Feb 2021

PONE-D-20-06089R1 

Implementing a new clinical pathway in a non-receptive context: mixed methods evaluation of a new fracture pathway for older people in a hospital Trust in the West Midlands, UK 

Dear Dr. Damery:

I'm pleased to inform you that your manuscript has been deemed suitable for publication in PLOS ONE. Congratulations! Your manuscript is now with our production department. 

Kind regards, 

on behalf of

Dr. Kathleen Finlayson 

Academic Editor

PLOS ONE